# Cold Tolerance and Population Dynamics of *Leptoglossus zonatus* (Hemiptera: Coreidae)

**DOI:** 10.3390/insects10100351

**Published:** 2019-10-17

**Authors:** Kristen E. Tollerup

**Affiliations:** University of California, Agriculture and Natural Resources, Kearney Agricultural Research and Extension Center, Parlier, CA 93648, USA; ketollerup@ucanr.edu

**Keywords:** leaffooted bug, truebug, almond, California’s San Joaquin Valley, pomegranate, overwintering, egg production

## Abstract

In California’s San Joaquin Valley, feeding by the coreid pest, *Leptoglossus zonatus*, can cause considerable economic loss on almond and pistachio. This research was conducted to improve understanding of how winter temperatures affect mortality of overwintering adult *L. zonatus* and to develop a better understanding of the role pomegranate plays in the species’ life-history. We exposed 7410 field-collected adult *L. zonatus* to temperatures between −2 and −10 °C for a period of three, four, or six hours using insect incubators. At six hours of exposure, the, LD_50_ and LD_95_ occur at −5.8 and −9.7 °C, respectively. We classified *L. zonatus* as chill-intolerant. Temperatures cold enough to affect substantial mortality of overwintering *L. zonatus* rarely occur in the San Joaquin Valley. Whole aggregation destructive sampling from a pomegranate hedgerow in Fresno County was conducted to determine population dynamics. At late summer to early fall, aggregations consisted of >90% immature stages. By early to mid-winter, mean aggregation size decreased, consisting of only three to 12 late-instars and adults. During years one and two of the experiment, *L. zonatus* produced a generation on pomegranate, mostly between September and mid-November. Overwintering did not occur on pomegranate, rather the majority of adults emigrated to other overwintering locations by mid-winter.

## 1. Introduction

Three species of leaffooted bug: *Leptoglossus clypealis* L. [1], *Leptoglossus occidentalis* Heidemann [2], and *Leptoglossus zonatus* (Dallas) [3] have been identified as agricultural pests in California’s San Joaquin Valley. The San Joaquin Valley covers approximately 42,000 km^2^ and extends approximately 720 km from the Sacramento, San Joaquin River Delta in the north, to the Tehachapi Mountains in the south. *Leptoglossus spp.* are polyphagous and feed on several crops important in the San Joaquin Valley such as: almond [4], citrus [5], corn [6], cotton, melons, pecan [7], pistachio [8], pomegranate [9], and tomato [10].

For reasons not completely understood, *L. zonatus* has become the predominate species in the San Joaquin Valley [11] and occurs throughout the valley, as well as the Sacramento Valley region. The native range of *L. zonatus* covers South and Central America; and North America, Mexico and the Southwest United States [12]. Although the species can feed on several crops produced in the San Joaquin Valley, economic loss has been confined primarily to almond and pistachio.

Interestingly, outside of California’s San Joaquin Valley, *L. zonatus* causes economic loss on a wider range of host crops. In Honduras for instance, the species causes serious economic loss of physic nut [13], and in Londrina Parana, Brazil, *L. zonatus* is known as a pest on sweet corn [14]. Moreover, in Louisiana, the species has caused serious damage on satsuma mandarin [15].

The economic loss to almond [4] and pistachio [3] results from direct and indirect damage to the developing nuts. Most loss results from early-season feeding on almond (March–May) and pistachio (May–June) which causes epicarp lesions and/or nuts aborting and dropping from trees. Feeding after May and June in almond and pistachio respectively, causes dark stains on the nut meat surface and/or kernel necrosis. Moreover, indirect damage results from the ability of *L. zonatus* to transmit the plant pathogen associated with panicle and shoot blight, *Neofusiccum mediterraneum* Crous, M.J. Wingf. and A. J. L. Phillip (formerly identified as *Botryosphaeria dothidae* Goss and Dug. De Not.) [16].

Rice et al. [1] reported that in late fall, adult leaffooted bugs form overwintering aggregations on evergreen plant species such as juniper and arborvitae as well as in leaflitter within protected areas of orchards. Seasonal dynamic studies [17,18] indicate that *L. zonatus* does not have a formal diapause. From late winter to early spring, depending on daytime temperatures, leaffooted bugs disperse from overwintering aggregations [17] into almond when nuts have developed to approximately one to two centimeters in diameter.

Pest control practitioners must rely on visual monitoring for the presence of adults and damaged almond nuts to detect the springtime arrival of the leaffooted bug [2]. Currently, pyrethroids remain the only class of insecticides that provide residual protection against the pest and account for the majority of applications. Mortality from cold winter temperatures and from natural enemies such as the egg parasitoid, *Gryon pennsylcanicum* (Ashmead) [1], and generalist predators, such as *Crysoperla* spp. *Zelus* spp., and ants (*Dorimyrmex*, *Solenopsis,* and *Formica* spp.) can have a negative impact on populations [19].

We conducted laboratory experiments to establish cold-temperature and exposure period thresholds of field-collected adult *L. zonatus*. This study also focused on elucidating the late-fall and early winter population dynamics of *L. zonatus* as well as the egg laying behavior and development of females in early spring. It is our goal that results of this research will help to advance the development of effective management tools.

## 2. Materials and Methods

### 2.1. Laboratory Colony

We established a colony from approximately 300 individuals, field-collected during late September 2014 from a commercial pomegranate orchard located in Fresno Co. Approximately 100 *L. zonatus* were placed in each of three 34.3 × 34.3 × 60 cm nylon cages (BioQuip products, Rancho Dominguez, CA, USA). Cages were kept in a climate-controlled greenhouse at a natural light photo period and approximately 20 to 30 °C. Colonies were provisioned with planters, raw redskin peanuts glued to strips of paper towel, green beans, *Phaseolus vulgaris* (L.) and water.

### 2.2. Cold Tolerance

During October and November (2015 and 2016); and October–December (2017), experiments were conducted to establish a low-temperature and temporal exposure threshold of adult *L. zonatus*. Field-collected leaffooted bug were contained in 34.3 × 34.3 × 60 cm nylon cages in the laboratory for no greater than seven days prior to their use in the experiment.

A single replication consisted of 10 individuals at approximately a 50:50 male / female ratio placed in an arena constructed from a 540 mL (FABRI-KAL Corp, Greenville, SC, USA) plastic food cup with a nylon screen placed over the top. Mortality was evaluated after exposure to temperature treatments of −10, −9, −6, −5, −2, and 0 °C (used as a control) for periods of three, four, or six hours in a low-temperature growth chamber (VWR, Radnor, Pennsylvania, USA). Temperature treatments were replicated at least six times. After being exposed to the cold treatments, the insects were removed from the low-temperature growth chamber and exposed to room temperature for 24 hours prior to evaluating mortality. Periodically during the experiment, low-temperature growth chambers were checked for temperature stability using an Onset HOBO Data Logger (Bourne, MA, USA).

### 2.3. Oviposition Period

To determine the oviposition period of *L. zonatus*, we collected three newly molted females from the laboratory colony and placed them in 3.785 L glass jars (Container and Packaging, Eagle, Idaho). In each jar, we provided an egg-laying substrate constructed from four, 24.7 cm long bamboo shish kabob skewers (HIC, Lakewood, New Jersey, NJ, USA) placed into a 4 × 4 cm^2^ block of FloraCraft dry foam (Ludington, Michigan, USA) at an angle of 30°. Two males were placed with the females until mating occurred, then removed. Due to a limitation of newly molted adult females we prepared six replications (jars) from 19 May to 15 June 2015. Jars were provisioned with the same food source as the colony and kept in the laboratory at 25 to 30 °C on a 16: 8 L: D photoperiod. Arenas were examined for egg deposition every one to three days, from 19 May to 1 September 2015.

### 2.4. Population Demographics

In mid-October 2016, a large population of *L. zonatus* was located on a ~1.6-km-long hedgerow of pomegranate near Reedley, Fresno Co. On October 19, two Hobo temperature data loggers were deployed within two pomegranate trees separated by approximately 90 m. Whole aggregations were collected by placing a sweep net bag over an entire branch on which the aggregation had formed and shaking the branch to dislodge the insects. The content of the sweep net bag then was placed into a 3.79-L, Ziploc plastic bag (SC Johnson and Son Inc, Racine, Wisconsin, USA) and labeled.

Four or five aggregations samples were taken in pomegranate on seven dates from 19 Oct. 2016 to 17 Jan. 2017, and 11 samples in 2017 from 26 September to 15 December. Aggregation samples were taken to the laboratory, placed in a freezer at approximately −17 °C until processed in no longer than 21 days. Collected *L. zonatus* were separated and enumerated according to developmental stage and sex. For the analyses, life stages were categorized by early (first, second, and third instar), late (fourth and fifth instar), and adult.

By 8 February 2016 and 24 January 2018 sample dates, no leaffooted bugs were on the pomegranate hedgerow. On the same dates, adult aggregations were located on Italian Cypress, *Cupressus sempervirens* (L.) and on four Mediterranean fan palm trees, *Chamaerops humilis* (L.) located approximately 30 to 50 m from the pomegranate hedgerow. The number of aggregated individuals on the Italian Cypress trees were counted. We sampled on three separate dates from early-February to mid-March 2016 and on three dates from late January to early March 2018. Visual counts, rather than destructive sampling, were used due to the aggregations consisting of a relatively small number of individuals and only the adult stage.

To determine when females begin producing eggs, we sub-sampled from the pomegranate hedgerow aggregations collected on five dates between 19 October 2016 and 17 March 2017. Females were dissected beneath a binocular dissecting scope in saline solution; the dorsal tergites were removed and the female reproductive system isolated as described in Sauza et al. [20]. If present, eggs were easily visible (Figure 1) and were counted.

### 2.5. Statistical Analyses

A general linear model procedure based on SAS® software 9.4 [21] was used to analyze the overall effect of temperature within each of the three exposure periods Prior to analysis, mortality data at each of the temperatures evaluated were corrected for control mortality using the Sun-Shepard’s formula [22]. Additionally, the general linear model procedure was utilized to evaluate differences among *L. zonatus* life stages, sex, and sex by date interaction. In all analyses, the least squares means procedure, along with the pdiff option, was utilized to determine statistical differences among means. Where appropriate, proportion data were arcsine square root transformed prior to the analysis. 

To develop a model predicting a 50% (LT_50_) and (LT_95_) 95% lethal temperature at each exposure period, data were evaluated using the probit procedure based on SAS software. The lack-fit procedure was used to test goodness-of-fit of the probit models at each exposure period.

## 3. Results

### 3.1. Cold Tolerance

From the fall of 2015 to early winter of 2017, a total of 7,410 field-collected adult *L. zonatus* were used in the analyses. No significant difference occurred between sexes and for each exposure period evaluated temperature data were pooled across all experiment dates. Control mortality at 0 °C reached a (mean ± SE) of 2.6 ± 0.7, 3.1 ± 0.7, and 2.6 ± 0.6% across the exposure periods of three, four, and six hours respectively. At the three-hour exposure treatment, percent mortality did not significantly increase above that of the control until −5°C (16 ± 3) (Table 1). A decrease of three degrees from −6 to −9 °C significantly increased mortality by approximately 1.4-fold, from 24 ± 3 to 58 ± 4%. Mortality at −10 °C reached 65 ± 5%, although it did not differ significantly from mortality at −9 °C. The generated probit model predicted that at three hours of exposure, the LT_50_ and LT_95_ occur at −8.4 and −12.8 °C respectively (Figure 2).

Mortality at the four-hour exposure period significantly increased above that of the control for all temperatures evaluated (Table 1). At −2 and −5 °C, mortality reached 23 ± 4 and 13 ± 3%, representing a 6.8 and 4.4-fold increase above control mortality, respectively. The difference between −2 and −5 °C did not significantly differ. Mean mortality of 48 ± 4% occurred at −6°C and significantly increased by approximately 0.4-fold from −6 to −9 °C and −6 to −10 °C (Table 1). At four hours of exposure, the probit model predicted the LT_50_, and LT_95_ to occur at −7.3 and −12.1 °C, respectively (Figure 2).

Six hours of exposure resulted in significantly greater mean mortality than the control at each temperature evaluated. A decrease of one degree, from −5 to −6 °C resulted in significantly higher mortality from 23 ± 3 to 76 ± 2%. A lower temperature from −6 to −10 °C resulted in a significant increase in percent mortality to 93 ± 2 (Table 1). No significant increase in mortality occurred from −6 to −9 °C. The probit model for the six-hour exposure period predicted the LT_50_ to occur at −5.8 °C and the LT_95_ to occur at −9.7 °C (Figure 2).

Over the two-year experiment, we aimed to collect field data to validate laboratory cold tolerant experiments. Whole aggregation samples were taken on 19 and 20 December 2016, before and after a mean low of −1.3 °C which occurred over a four-hour period. The mean percentage of adults per aggregation differed numerically from 81 ± 6 on 19 December to 78 ± 10 after the freeze event (Table 1), although the decrease was not statistically significant. Additionally, no significant difference occurred in the percentage mortality of early or mid-instar stages due to the freeze event (Figure 3).

### 3.2. Oviposition Period

Females in six arenas were observed over a period of (mean ± SE) 82.3 ± 5.0 days. In a single arena, one of the three females died prior to the beginning of egg-laying. Egg-laying began after 23.7 ± 6.4 days and continued for 58.7 ± 1.4 days. At the initiation of egg-laying females laid a mean of 13.0 ± 2.2 eggs with a cumulative mean of 72.2 ± 7.2 eggs.

### 3.3. Population Dynamics

Aggregations on the pomegranate hedgerow were sampled from 19 October 2016 until mid-January 2017 and from 26 September 2017 to 15 December 2017 (Figure 3). Based on egg to adult development time of (mean ± SE) 44.6 ± 1 days at 30 °C (Jackson et al 1995), we determined that *L. zonatus* began laying eggs in early to mid-September during both 2016 and 2017.

Over sampling dates on the pomegranate hedgerow, mean aggregation size ranged from 206.2 ± 39.7 to 2.8 ± 0.20 in 2016 and 76.3 ± 24.3 to 11.8 ± 1.8 during 2017. On 8 February 2017 and 24 January 2018 aggregations were not found on pomegranate, therefore, we began monitoring on twenty Cypress trees located approximately 30 to 50 m from the pomegranate hedgerow (Figure 3).

In early February 2017, aggregations on Cypress consisted of only the adult stage and a mean of 31 ± 9.1 individuals. At early March aggregations began to disperse and by 15 March consisted of only 1.8 ± 0.58 individuals (Figure 3). In 2018, samples were collected from Cypress on three dates and consisted of only the adult stage. Mean aggregation size was 12.2 ± 4.4 in late January and 4.2 ± 1.02 by early March (Figure 3). By late March of both years, aggregations on Cypress were absent.

A decreasing percentage of early-stage instars (first, second, and third) occurred from 19 October (45 ± 8) to 17 January 2017 (0); and 26 September (87 ± 6) to 15 December 2017 (5 ± 2). Late-stage instars (fourth and fifths) initially consisted of approximately 47 ± 5% of individuals in aggregations during 2016 sample dates, and similar to early instars, the percentage decreased to approximately 13 by mid-January. In late September 2017, the mean percentage of late instar *L. zonatus* equaled 10 ± 5 and remained above 24 until 15 December (Figure 3).

In the first year of the experiment, mean adults per aggregation on the pomegranate hedgerow at the initiation of sampling was below nine percent until early November 2016. Adults per aggregation increased between 7 and 30 November 2016 from 8 ± 2 to 66 ± 5% respectively and remained high, above 78%, on subsequent sampling dates. During the second year, a similar pattern occurred with adults per aggregation remaining below nine percent until early October 2017. On subsequent sampling dates, the percentage of adults increased from 15 ± 4 at mid-October to 71 ± 11 by 15 December 2017 (Figure 3).

Mean percentage of adult females per aggregation at late-fall on pomegranate equaled 44 ± 7 (2016) and 38 ± 5 (2017). At early spring, females per aggregation on Cypress equaled 39 ± 2 (2016) and 48 ± 2% (2018). Aggregations on each of the sample dates had a significantly greater number of males (F = 12.25; df = 1; *p* < 0.01). No significant interaction of date by sex occurred (F = 0.98; df = 3; *p* > 0.05).

On late-fall and winter sample dates (Table 2), no dissected females contained eggs. We first began observing eggs in dissected females from two sample dates, 22 February and 3 March 2017. Approximately 26% contained a mean of 5.6 ± 7.0 eggs per female. On the last sample date, just prior to *L. zonatus* leaving overwintering aggregations, approximately 42% of dissected females contained a mean of 16 ± 2.0 eggs. 

## 4. Discussion

Survival of overwintering insects depends on the ability to avoid intracellular freeze damage by supercooling [23]. All insects utilize supercooling, at least to some degree, through the accumulation of cryoprotectant compounds in the hemolymphs such as low molecular weight alcohols e.g. glycerol [24] and/or large protein molecules [25]. Broadly speaking, three categories of insect cold tolerance classifications exist: freeze-tolerant, species that can survive some freezing; freeze-intolerant, species that die upon freezing; and chill-intolerant, insects that suffer mortality prior to freezing [26]. In this experiment, we found that mortality of adult *L. zonatus* occurred at temperatures above their point of freezing. Laboratory experiments confirmed our hypothesis that *L. zonatus* can be classified as a chill-intolerant species.

Results of our cold tolerance study align with previous studies of other insect species. Cira et al. [27] found that mortality of brown marmorated stink bug, *Halyomorpha halys* (Stål) occurred at temperatures significantly warmer than its freezing point. From populations collected in Minnesota and Virginia, the group determined the supercooling point of *H. halys* at approximately −15 and −13.9 °C respectively. Moreover, they found that approximately 18 to 24% mortality of *H. halys* occurred when the insect was cooled to −5 °C. Additionally, in the redbanded stink bug, *Piezodorus guildinii* (Westwood), supercooling occurs at approximately −11 °C [28]. Similar to *L. zonatus*, *P. guildinii* suffers considerable mortality prior to freezing, with a 50% lethal exposure period of 6.8 hours at −5 °C.

The predicted lethal temperatures, LT_50_ and LT_95_, of overwintering *L. zonatus* indicate that the species is cold-tolerant and before substantial mortality occurs, temperatures must decrease to lows that seldom occur in California’s Central Valley. Between 2003 and 2019, for instance, minimum temperatures from −5 to −6 °C occurred during only 2007, 2012, and 2013. Our model developed for six hours of exposure is supported by observations made by Daane et al. [17] during the winter of 2007 when the temperature dropped to approximately −6 °C for six hours and substantial mortality of adult and immature *L. zonatus* occurred.

Based on our probit model for the four-hour exposure period, we expected to find approximately 15 to 20% mortality of adult *L. zonatus* due to the cold temperatures reached between 19 and 20 December 2016. The lack of mortality we observed may be due to a couple of factors. First, aggregations can generate a microclimate more favorable to survival through water conservation [29]. Second, during early December 2016, aggregations were periodically exposed to cold ambient temperatures. Such exposure can increase cold hardiness [30]. Our model validation data suggest that the *L. zonatus* evaluated in the laboratory may have been slightly less cold-hardy than the population evaluated on the pomegranate hedgerow. Our cold tolerance models, therefore may slightly overestimate mortality.

Sampling *L. zonatus* on the pomegranate hedgerow during 2016 and 2017 indicate that the species produced a full generation between early September to approximately mid-November. Jackson et al. [31] established development temperature parameters for *L. zonatus* life stages, however, they did not determine the number of degree day units needed to develop from egg to adult. In the closely related leaffooted bug, *L. occidentalis*, Barta [32], determine that the species required 533 DD°C from egg to adult. This closely aligns to the number of degree day units that occurred between early September and mid-November in both years of this study. 

Upon reaching the adult stage, all individuals emigrated from the pomegranate hedgerow to a more sheltered location rather than remaining in the tree canopy or under trees beneath leaf litter. It is probable that some of the adults emigrated to the Cypress trees given their proximity to the pomegranate hedgerow. This is in contrast to previous reports of *L. zonatus* remaining in pomegranate throughout the winter [2]. In a closely related species, researchers have reported overwintering aggregations of adult *L. clypealis* on plant and non-plant substrates such as: juniper and arborvitae [1], beneath tree bark, in protected areas of orchards in leaf litter [33]; and in and around barns and field irrigation pump houses [1].

In aggregations on both the pomegranate hedgerow and Cypress trees we found a (F:M) sex ratio significantly skewed to males. Our findings contrast that of other investigators [18,32] who found a ratio very close to 1:1 in *L. zonatus* during developmental trials. Our results may have occurred due to the sampled aggregations consisting of two overlapping generations rather than a single cohort of new adults.

The reproduction biology of *L. zonatus* has been well studied [6,15,18,31]. However, we are not aware of any published studies that determined at what point in the season *L. zonatus* females contain developed eggs. Our findings determined that *L. zonatus* develop eggs at late-winter to early-spring, which coincides with the period the species begins leaving overwintering aggregations. Also, in laboratory experiments, we found that the oviposition period of *L. zonatus* lasts for ~59 days. On satsuma mandarin, the oviposition period of *L. zonatus* was considerably shorter, only approximately 13 days, while the range, however, was 12 to 64 days. Daane et al. [18] found an oviposition period of 42 days when fed on a diet of green beans and Cypress.

## 5. Conclusions

*Leptoglossus zonatus* stands as a key coreid pest of almond, as well as pistachio in California’s San Joaquin Valley. Almond provides an abundant food source for *L. zonatus* as they emigrate from overwintering sites during early spring. Almond production in the San Joaquin Valley represents a multi-billion-dollar industry and consists of approximately 0.48 million bearing and non-bearing hectares. We report that pomegranate plays a key role in providing a suitable host for the development of a third overwintering generation. Commercial pomegranate production occurs on ~14,300 hectares, planted mostly in the southern regions of the valley. Moreover, throughout the valley, farmers and homeowners commonly utilize pomegranate as ornamentals, living fences, and hedgerows because of the plant’s aesthetics, dense canopy, and tolerance to drought. Providing growers and pest control practitioners with a better understanding of pomegranate’s role in the life-history of *L. zonatus* can help improve monitoring and management practices, as well as predicting spring populations. Furthermore, we report that for overwintering *L. zonatus* to suffer substantial mortality, temperatures must fall to lows not common in California’s San Joaquin Valley.

## Figures and Tables

**Figure 1 insects-10-00351-f001:**
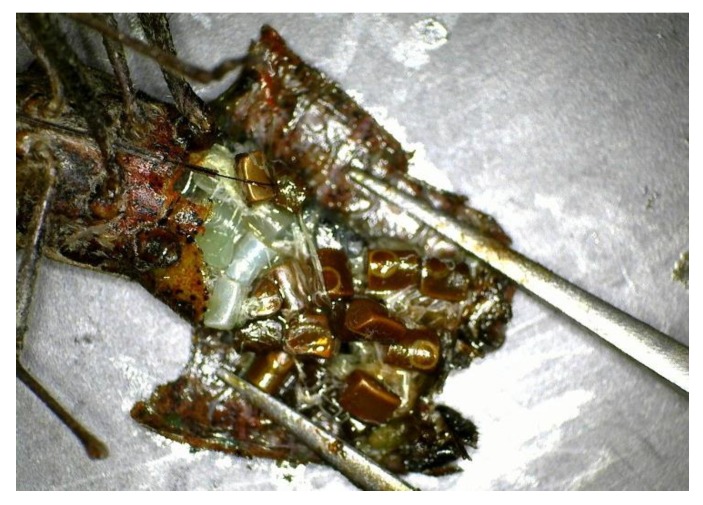
Dissected female with developed eggs.

**Figure 2 insects-10-00351-f002:**
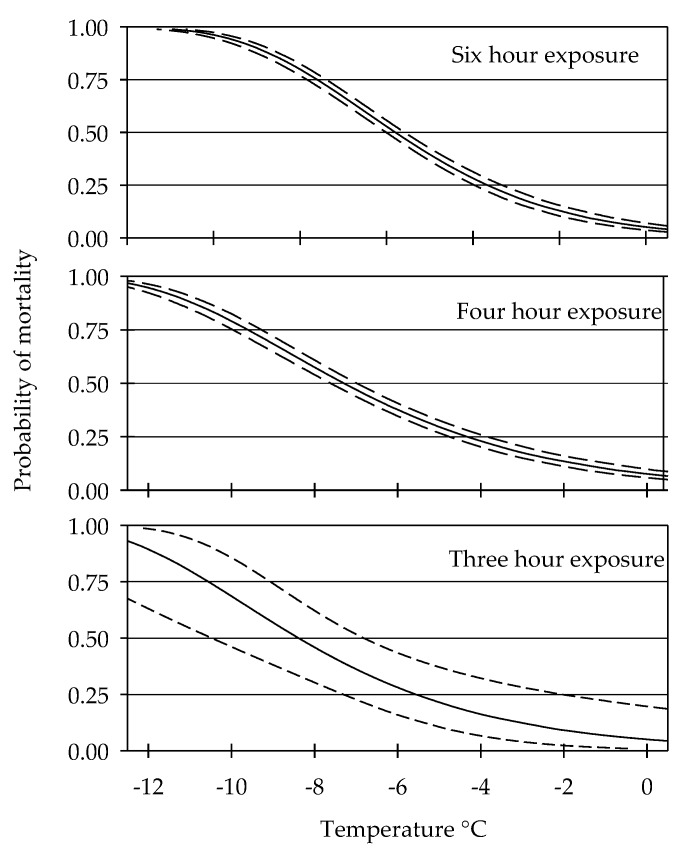
Probit models for exposure to three, four, and six hours of sub-freezing temperature.

**Figure 3 insects-10-00351-f003:**
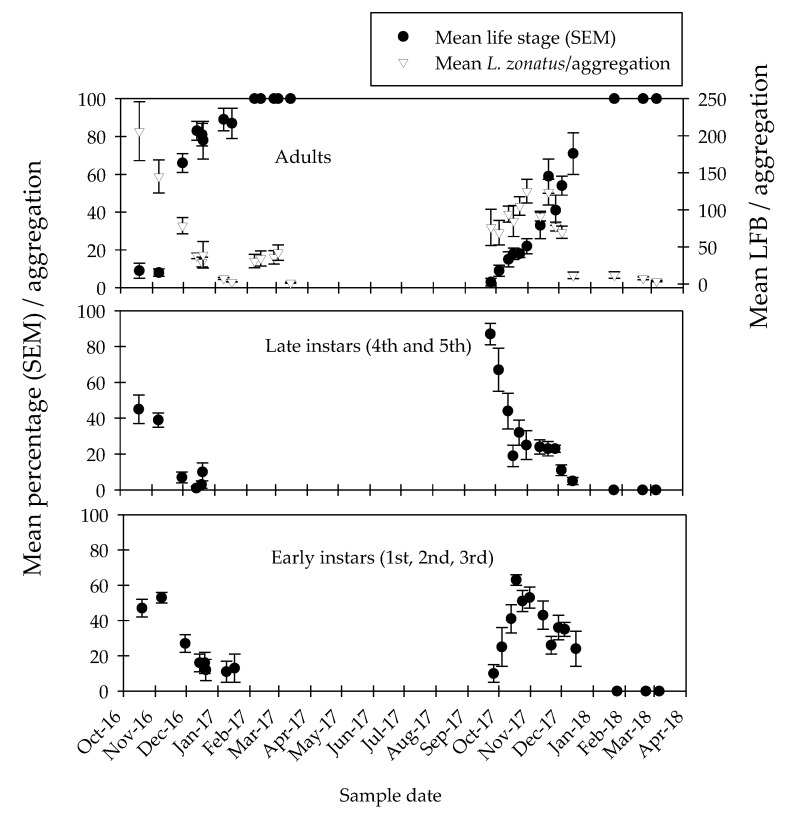
Mean number of individuals and percentage of early, late, and adult *L. zonatus* per aggregation sampled. Aggregations collected on pomegranate (19 Oct 2016 to 17 Jan 2017, and 26 Sept 2017 to 15 Dec 2017). Aggregations collected on Cypress (8 Feb 2017 to 15 Mar 2017, and 24 Jan 2018 to 6 Mar 2018).

**Table 1 insects-10-00351-t001:** Mean percentage mortality of adult leaffooted bug after exposure to sub-freezing temperatures for three different exposure periods.

	Hours of Exposure					
Temperature	3		4		6	
°C	n	mean ± (SE)	n	mean ± (SE)	n	mean ± (SE)
−10	180	^*^ 65 ± 5 a	260	70 ± 4 a	240	93 ± 2 a
−9	120	58 ± 4 a	120	67 ± 4 a	120	80 ± 4 b
−6	480	24 ± 3 b	480	48 ± 4 b	480	76 ± 2 b
−5	540	16 ± 3 c	660	13 ± 3 c	840	23 ± 3 c
−2	120	11 ± 3 cd	60	23 ± 4 c	60	25 ± 3 c
0	780	2.6 ± 0.7 d	850	3.1 ± 0.7 d	1020	2.6 ± 0.6 d

*^*^* Means within columns followed by the same letter are not significantly different.

**Table 2 insects-10-00351-t002:** Mean percentage of dissected females with eggs and eggs per female sub-sampled from aggregations collected during winter and spring.

Date	Number of Samples	Total Dissected	Mean Percentage w/Eggs	Mean Eggs/Female
19 Oct–19 Dec 2016	10	71	0	0
22 Feb–3 Mar 2017	8	45	26 ± 1.3	5.6 ± 7.0
17–Mar 2017	11	53	42 ± 0.8	16 ± 2.0

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
