# Peer review of "Cold Tolerance and Population Dynamics of Leptoglossus zonatus (Hemiptera: Coreidae)"

_insects, 2019, doi:10.3390/insects10100351_

Round 1

Reviewer 1 Report

This manuscript reports cold tolerance, oviposition period and population demographics during overwintering in Leptoglossus zonatus. Because this insect is an agricultural pest of several crops in California, physiological information on cold tolerance and their demography can contribute future pest management. These results are valuable. However, several problems mentioned below should be considered.

This manuscript did not mention diapause, although diapause is generally an important mechanism for overwintering insects. Does not L. zonatus show diapause? If so, this information should be mentioned.

The authors report that L. zonatus migrated from pomegranate to Cypress trees during winter. I think that this is an interesting phenomenon. However, another scenario is possible. The insects observed in late winter on Cypress trees may had arrived in fall season. Discussion or additional observation is needed if this scenario is possible.

At the line 254, the authors discuss the difference between mortality predicted and observed. However, potential effects of acclimation are neglected. The experiment on cold tolerance was conducted for wild individuals collected in the fall season. They may be weaker for the cold temperatures than wild individuals observed during winter.

The method of statistical analysis is often unclear in this manuscript. For instance, the mortality rate of adults is compared in Table 1. The values with various letters hints that a multiple comparison is conducted. However, this procedure is not explained. Furthermore, the significance level should be described in Table 1.

At the lines 206-210, the results of statistical analysis on sex ratio of aggregation are shown. However, the method of analysis is unclear. I think that the materials and methods do not explain this analysis. This paragraph should be revised.

The point of freezing is a key word in the manuscript. Therefore, the value of freezing point in L. zonatus should be described.

Author Response

Reviewer 1

Comment 1: This manuscript did not mention diapause, although diapause is generally an important mechanism for overwintering insects. Does not L. zonatus show diapause? If so, this information should be mentioned.

Reply: There is no literature that explicitly address the issue of diapause of L. zonatus.  In a personal communication with Kent Daane, he indicated that L. zonatus could be collected from the field in winter, brought into a laboratory setting and the insects will nearly immediately begin feeding and mating.   I have to the text to clarify.

Comment 2: The authors report that L. zonatus migrated from pomegranate to Cypress trees during winter. I think that this is an interesting phenomenon. However, another scenario is possible. The insects observed in late winter on Cypress trees may had arrived in fall season. Discussion or additional observation is needed if this scenario is possible.

Reply: Yes, it is possible that L. zonatus adults arrived earlier.  The primary point that I want to convey is that the all adults emigrated from the pomegranate.  It is highly probable that the many adults emigrated to the Cypress given that it is a known overwintering location and the trees’ proximity to the pomegranate hedgerow.

Comment 3: At the line 254, the authors discuss the difference between mortality predicted and observed. However, potential effects of acclimation are neglected. The experiment on cold tolerance was conducted for wild individuals collected in the fall season. They may be weaker for the cold temperatures than wild individuals observed during winter.

Reply: Reviewer 1 expressed a concern that the individuals tested in the cold tolerance experiments may not be as cold hardy as the population observed on the pomegranate.  I eluded to this possibility.  I have rewritten the statements in the materials and methods and the results sections in an effort to clarify. 

Comments 4 and 5: The method of statistical analysis is often unclear in this manuscript. For instance, the mortality rate of adults is compared in Table 1. The values with various letters hints that a multiple comparison is conducted. However, this procedure is not explained. Furthermore, the significance level should be described in Table 1.

At the lines 206-210, the results of statistical analysis on sex ratio of aggregation are shown. However, the method of analysis is unclear. I think that the materials and methods do not explain this analysis. This paragraph should be revised.

Reply:  I have edited the statistical analyses section to clarify the methods used to separate means of mortality among the cold exposure periods Table 1, and the sex percentage means.

Comment 6: The point of freezing is a key word in the manuscript. Therefore, the value of freezing point in L. zonatus should be described.

We do have data (not shown in the manuscript) that strongly suggest that L. zonatus freezes close to -17°C.  This is not included in the manuscript since it would be too speculative.   

Reviewer 2 Report

Accurate understanding details of life cycle and biology (e.g. how winter temperatures affect mortality of overwintering specimens) is necessary in order to better predict the seasonal occurrence and abundance of an insect pest and finally allow protection of crops against them.

This manuscript is an original research article presents the results of research on an important pest of almond and pistachio - Leptoglossus zonatus and responds to the need mentioned above

The manuscript fulfils the requirements of the "Insects" journal in all aspects.

I have only a few minor, technical comments (that I marked in the manuscript.

Author Response

there were very few comments from reviewer 2.
